# Differentially Private Multi-Party Data Release for Linear Regression

**Ruihan Wu**[*1]   **Xin Yang**[2]   **Yuanshun Yao**[2]   **Jiankai Sun**[2]   **Tianyi Liu**[2]   **Kilian Q. Weinberger**[1]   **Chong Wang**[2]

[1]Cornell University, USA
[2]ByteDance Inc., USA
[*]Work done as an intern at ByteDance Inc.

## Abstract

*Differentially Private (DP) data release* is a promising technique to disseminate data without compromising the privacy of data subjects. However the majority of prior work has focused on scenarios where a single party owns all the data. In this paper we focus on the multi-party setting, where different stakeholders own disjoint sets of attributes belonging to the same group of data subjects. Within the context of linear regression that allow all parties to train models on the complete data without the ability to infer private attributes or identities of individuals, we start with directly applying Gaussian mechanism and show it has the small eigenvalue problem. We further propose our novel method and prove it asymptotically converges to the optimal (non-private) solutions with increasing dataset size. We substantiate the theoretical results through experiments on both artificial and real-world datasets.

## 1   INTRODUCTION

The machine learning community has greatly benefited from open and public datasets [Chapelle and Chang, 2011, Real et al., 2017, Fast and Horvitz, 2017, Kong et al., 2020]. Unfortunately the privacy concern of data release significantly limits the feasibility of sharing many rich and useful datasets to the public, especially in privacy-sensitive domains like health care, finance, and government etc. This restriction considerably slows down the research in those areas as well as the general machine learning research given many of today's algorithms are data-hungry. Recently, legal and moral concerns on protecting individual privacy become even greater. Most countries have imposed strict regulations on the usage and release of sensitive data, e.g. CCPA [Legislature, 2018], HIPPA [Act, 1996] and GDPR [Parliament and of the European Union, 2016]. The tension between

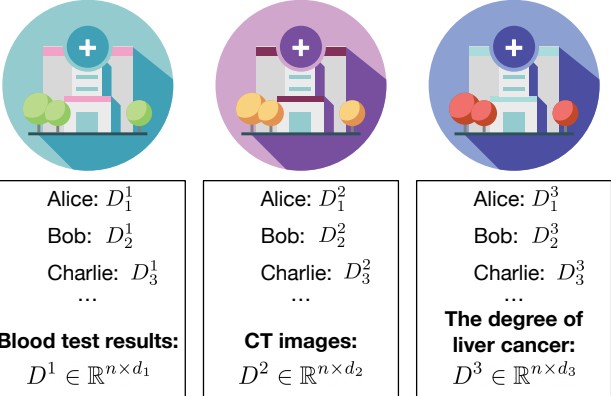

Figure 1: An illustration of how data is distributed in the health care example. Clinics have the same set of patients, but different attributes such as blood test results, CT images and the degree of liver cancer.

protecting privacy and promoting research drives the community as well as many ML practitioners into a dilemma.

*Differential privacy* (DP) [Dwork, 2011, Dwork et al., 2006, 2014, Sheffet, 2017, Lee et al., 2019, Xu et al., 2017, Kenthapadi et al., 2012] is shown to be a promising direction to release datasets while protecting individual privacy. DP provides a formal definition of privacy to regulate the trade-off between two conflicting goals: protecting sensitive information and maintaining data utility. In a DP data release mechanism, the shared dataset is a function of the aggregate of all private samples and the DP guarantees regulate how difficult for anyone to infer the attributes or identity of any individual sample. With high probability, the public data would be barely affected if any single sample were replaced.

Despite the ongoing progress of DP data release, the majority of the prior work mainly focuses on the single-party setting which assumes there is only one party that would release datasets to the public. However in many real-world scenarios, there exist multiple parties who own data relevant to each other and want to collectively share the data as a

*Accepted for the 38th Conference on Uncertainty in Artificial Intelligence* (UAI 2022).

whole to the public. For example, in health care domain, some patients may visit multiple clinics for specialized treatments (Figure 1), and each clinic only has access to its own attributes (e.g. blood test and CT images) collected from the patients. For the same set of patients, attributes combined from all clinics can be more useful to train models. In general, the multi-party setting assumes multiple parties own disjoint sets of attributes (features or labels) belonging to the same group of data subjects (e.g. patients).

One straightforward approach to release data in a multi-party setting is combining data from all parties in a centralized place (e.g. one of the data owners or a third-party), and then releasing it using a private single-party data release approach. However, in a privacy-sensitive organization like a clinic, sending data to another party is prohibited by policy. An alternative approach is to let each party individually release its own data to the public through adding sample-wise Gaussian noise, and then ML practitioners can combine the data together to train models. However the resulting models trained on the data combined in this way would show a significantly lower utility compared to the models trained on non-private data (confirmed by experiments in Section 5). To bridge this utility gap, we propose new algorithms specifically designed for multi-party setting.

In summary, we study DP data release in multi-party setting where parties share attributes of the same data subjects publicly through a DP mechanism. It protects the privacy of all data subjects and can be accessed by the public, including any party involved. To this end, we propose the following two differentially private algorithms, both based on Gaussian DP Mechanism [Dwork et al., 2014] within the context of linear regression. First, in *De-biased Gaussian Mechanism for Ordinary Least Squares (DGM-OLS)*, each party adds Gaussian noise directly to its data. The learner with the public data is able to remove a calculated bias from the Hessian matrix. However, we show that bias removal brings the small eigenvalue problem. Hence, we propose the second method *Random Mixing prior to Gaussian Mechanism for Ordinary Least Squares (RMGM-OLS)*. A random Bernoulli projection matrix is shared to all parties, and each party uses it to project its data along sample-wise dimension before adding Gaussian noise. We prove that both algorithms are guaranteed to produce solutions that asymptotically converge to the optimal solutions (i.e. non-private) as the dataset size increases. Through extensive experiments on both synthetic and real-world datasets, we show the latter method achieves the theoretical claims and outperforms the first method that naively adapts Gaussian mechanism.

## 2 PRELIMINARY

A sequence $\{X_n\}$ of random variables in $\mathbb{R}^d$ is defined to *converge in probability* towards the random variable $X$ if

for all $\beta > 0$,
$$\lim_{n \to \infty} \mathbb{P}\left[\|X_n - X\| > \beta\right] = 0.$$

The norm notation $\|\cdot\|$ denotes $\ell^2$ norm in our paper. We denote this convergence as $\text{plim}_{n\to\infty} X_n = X$.

*Differential privacy* (DP; [Dwork et al., 2006, 2014]) is a quantifiable and rigorous privacy framework, which is formally defined as follows.

**Definition 1** (($\varepsilon, \delta$)-differential privacy). *A randomized mechanism $\mathcal{M} : \mathcal{D} \to \mathcal{R}$ with domain $\mathcal{D}$ and range $\mathcal{R}$ satisfies $(\varepsilon, \delta)$-differential privacy if for any two adjacent datasets $D, D' \in \mathcal{D}$, which differ at exactly one data point, and for any subset of outputs $S \subseteq \mathcal{R}$, it holds that*

$$\mathbb{P}[\mathcal{M}(D) \in S] \le e^{\varepsilon} \cdot \mathbb{P}[\mathcal{M}(D') \in S] + \delta.$$

*Gaussian mechanism* [Dwork et al., 2014] is a post-hoc mechanism to convert a deterministic real-valued function $f : \mathcal{D} \to \mathbb{R}^m$ to a randomized algorithm with differential privacy guarantee. It relies on *sensitivity* of $f$, denoted by $S_f$, which is defined as the maximum difference of output $\|f(D) - f(D')\|$. We define Gaussian mechanism for differential privacy as below.

**Lemma 1** (Gaussian mechanism). *For any deterministic real-valued function $f : \mathcal{D} \to \mathbb{R}^m$ with sensitivity $S_f$, we can define a randomized function by adding Gaussian noise to $f$:*

$$f^{dp}(D) := f(D) + R,$$

*where R is sampled from a multivariate normal distribution $\mathcal{N}\left(\mathbf{0}, S_f^2 \sigma^2 \cdot I\right)$. When $\sigma \ge \frac{\sqrt{2\log(1.25/\delta)}}{\varepsilon}$, $f^{dp}$ is $(\varepsilon, \delta)$-differentially private for $0 < \varepsilon \le 1$ and $\delta > 0$.*

To simplify notations, we define $\sigma_{\varepsilon,\delta} := \frac{\sqrt{2\log(1.25/\delta)}}{\varepsilon}$.

*Johnson-Lindenstrauss lemma* (JL; [Johnson and Lindenstrauss, 1984, Achlioptas, 2003]) is a technique to compress a set of vectors $S = \{v_1, \cdots, v_l\}$ with dimension $d$ to a lower dimension space $k < d$. With a proper selection $k$, it is able to approximately preserve the inner product between any two vectors in the set $S$ with high probability. We specifically introduce the Bernoulli version of JL Lemma, which is extended from Theorem 1.1 in Achlioptas [2003].

**Lemma 2** (JL Lemma for inner-product preserving (Bernoulli)). *Suppose $S$ is an arbitrary set of $l$ points in $\mathbb{R}^d$ and suppose $s$ is an upper bound for the maximum $\ell^2$-norm for vectors in $S$. Let $B$ be a $k \times d$ random matrix, where $B_{ij}$ are independent random variables taking value from $1$ or $-1$ with probability $1/2$ respectively. With the probability at least $1 - (l+1)^2 \exp\left(-k\left(\frac{\beta^2}{4} - \frac{\beta^3}{6}\right)\right), \forall \mathbf{u}, \mathbf{v} \in S$, we have*

$$\frac{\mathbf{u}^\top \mathbf{v}}{s^2} - 4\beta \le \frac{\left(B\mathbf{u}/\sqrt{k}\right)^\top \left(B\mathbf{v}/\sqrt{k}\right)}{s^2} \le \frac{\mathbf{u}^\top \mathbf{v}}{s^2} + 4\beta.$$

# 3 NOTATION AND PROBLEM SETUP

**Notations.** Denote $D^j$, $j = 1, \cdots, m$, as data matrices for $m$ parties, where $D^j \in \mathbb{R}^{n \times d_j}$ and $m \geq 2$. They are aligned by the same set of subjects but have different attributes and they have the same number of samples. Define $D = \left[ D^1, \cdots, D^m \right] \in \mathbb{R}^{n \times (d+1)}$ as the collection of all datasets, where $d = d_1 + \cdots + d_m - 1$. We define $d$ by subtracting 1 from the total number of attributes because one column is label which we need to treat separately. Define $d_{\max} = \max_{j \in [m]} d_j$, and $D_i$ as the $i$-th row of $D$, we make the following assumption on data distribution:

**Assumption 1.** *$D_i$, $i = 1, \cdots, n$, are i.i.d sampled from an underlying distribution $\mathcal{P}$ over $\mathbb{R}^{d+1}$.*

**Dataset release algorithm.** A private multi-party data release algorithm needs to protect both inter-party and intra-party communications. The general workflow of our proposed algorithms is designed as the following:

1. Pre-generate random variable $B$. The pre-generated one or more random variables will be shared among parties.
2. Privatize the dataset locally with the algorithm $\mathcal{A}^{\mathrm{priv}}$. Each party applies the same privatizing algorithm $\mathcal{A}^{\mathrm{priv}}$ that takes the local dataset $D^j \in \mathbb{R}^{n \times d_j}$ and the random matrix $B$ as the inputs and then outputs $k$ (predefined) "encrypted" samples $\left( D^{\mathrm{pub}} \right)^j := \mathcal{A}^{\mathrm{priv}}(D^j; B) \in \mathbb{R}^{k \times d_j}$.
3. Release the dataset. All parties jointly release $D^{\mathrm{pub}} = \left[ \left( D^{\mathrm{pub}} \right)^1, \cdots, \left( D^{\mathrm{pub}} \right)^m \right] \in \mathbb{R}^{k \times (d+1)}$ to the public.

Note that we need to specially design random variable $B$ and the privatizing algorithm $\mathcal{A}^{\mathrm{priv}}$, which we will introduce in the next section. In addition, the random variable $B$ allows the dependencies between the randomized output from all parties, which can be utilized to guarantee the final utility.

**Privacy constraint.** Since the public will observe the released dataset $D^{\mathrm{pub}}$, for each $j \in [m]$, $\left( D^{\mathrm{pub}} \right)^j$ should not leak the information of the private dataset $D^j$. Formally we require $\forall j \in [m]$, $\mathcal{A}^{\mathrm{priv}}(D^j; B)$ is differentially private, where two neighbouring datasets $D^j$ and $\left( D^j \right)'$ differ at one row (sample).

However the multi-party setting requires more than the above guarantee because each party $j' \neq j$ not only observes $D^j$ but also the shared random variable $B$. Thus we need to further require that given $B$, each party $j$ cannot infer information about other private datasets $D^j$. In terms of differential privacy, it is required that condition on $B$ for any possible sample value $I$, $\mathcal{A}^{\mathrm{priv}}(D^j; B)$ is $(\varepsilon, \delta)$-differentially private, *i.e.* for any two neighbouring datasets $D^j$ and $\left( D^j \right)'$ and $B$, we have

$$\mathbb{P}(\mathcal{A}^{\mathrm{priv}}(D^j; B)|B) \leq e^\varepsilon \cdot \mathbb{P}\left( \left( \mathcal{A}^{\mathrm{priv}}\left( \left( D^j \right)'; B \right) \middle| B \right) + \delta. \right.$$

**Utility target.** We aim to guarantee the performance of *arbitrary* linear regression task (arbitrarily selected label and features) on the joint released dataset $\left[ D^1, \cdots, D^m \right]$. Out of the notation simplicity, we assume the label in the linear regression task is the last attribute, and the features are the rest of the attributes. Under this assumption, the joint private dataset $D$ can be written as $[X, Y]$, where $X \in \mathbb{R}^{n \times d}$ is the private feature matrix and $Y \in \mathbb{R}^n$ is the private label vector. Similarly the public dataset $D^{\mathrm{pub}}$ can be written as $[X^{\mathrm{pub}}, Y^{\mathrm{pub}}]$, where $X^{\mathrm{pub}} \in \mathbb{R}^{k \times d}$ and $Y^{\mathrm{pub}} \in \mathbb{R}^k$.

We define the loss function by the expected squared loss:

$$L(\mathbf{w}; \mathcal{P}) = \mathbb{E}_{(\mathbf{x}, y) \sim \mathcal{P}} \left[ (\mathbf{w}^\top \mathbf{x} - y)^2 \right], \qquad (1)$$

where the data point is sampled from the distribution $\mathcal{P}$ in Assumption 1. We make two more assumptions for the distribution $\mathcal{P}$: the standard normalization and the *no perfect multicollinearity* assumption. The latter is common in the literature of linear regression [Farrar and Glauber, 1967, Chatterjee and Hadi, 2006].

**Assumption 2.** *The absolute values of all attributes $|D_{ij}|$ are bounded by 1.*

**Assumption 3.** *$\mathbb{E}_{(\mathbf{x}, y) \sim \mathcal{P}} \left[ \mathbf{x} \mathbf{x}^\top \right]$ is positive definite.*

Under Assumption 3, derived by setting $\nabla_{\mathbf{w}} L(\mathbf{w}; \mathcal{P}) = 0$, the optimal solution $\mathbf{w}^*$ to the loss in Equation 1 has the following explicit form:

$$\mathbf{w}^* = \left( \mathbb{E}_{(\mathbf{x}, y) \sim \mathcal{P}} \left[ \mathbf{x} \mathbf{x}^\top \right] \right)^{-1} \mathbb{E}_{(\mathbf{x}, y) \sim \mathcal{P}} \left[ \mathbf{x} \cdot y \right].$$

The utility target (for the trained linear regression model) is determined by our release algorithm $(B, \mathcal{A}^{\mathrm{priv}})$. For a given public dataset $D^{\mathrm{pub}}$ released by our algorithms, we define our utility target as the existence of a training algorithm $\mathcal{A}^{\mathrm{lr}}$ that achieves the asymptotic property for the trained model weights $\hat{\mathbf{w}}_n := \mathcal{A}^{\mathrm{lr}}\left( D^{\mathrm{pub}} \right)$ as the dataset size $n \to \infty$. The asymptotic property is commonly studied in differential privacy [Chaudhuri and Hsu, 2011, Bassily et al., 2014, Feldman et al., 2020] and we restate it as follows: $\hat{\mathbf{w}}_n$ converges to $\mathbf{w}^*$ in probability as the size of dataset $n$ increases, *i.e.* $\forall \beta > 0$, $\lim_{n \to \infty} \mathbb{P}\left[ \|\hat{\mathbf{w}}_n - \mathbf{w}^*\| > \beta \right] = 0$. The randomness from the above property comes from data sampling $\mathcal{P}$, dataset release algorithm $(B, \mathcal{A}^{\mathrm{priv}})$, and the training algorithm $\mathcal{A}^{\mathrm{lr}}$.

# 4 METHODOLOGY

We now describe our data release algorithms which both satisfy the differential privacy and yield asymptotically optimal solutions to the linear regression task. We start with the first algorithm *De-biased Gaussian Mechanism for Ordinary Least Squares (DGM-OLS)*, which directly applies Gaussian mechanism when releasing the data and then de-biases

**Algorithm 1** DGM-OLS

**Dataset Release**
1: **Input:** $D = [D^1, \cdots, D^m], \varepsilon, \delta$.
2: **for** $j = 1, \cdots, m$ **do**
3:   The party $j$ computes $(D^{\mathsf{dgm}})^j := D^j + R^j$, where $R^j \in \mathbb{R}^{n \times d_j}$ is a random Gaussian matrix and elements in $R^j$ are i.i.d sampled from $\mathcal{N}\left(0, 4d_{\max} \cdot \sigma_{\varepsilon,\delta}^2\right)$.
4: **end for**
5: **Return:** $D^{\mathsf{dgm}} =: \left[(D^{\mathsf{dgm}})^1, \cdots, (D^{\mathsf{dgm}})^m\right]$.

**Training Algorithm**
1: **Input:** $D^{\mathsf{dgm}}, \varepsilon, \delta$
2: $[X^{\mathsf{dgm}}, Y^{\mathsf{dgm}}] = D^{\mathsf{dgm}}$
3: Compute the de-biased Hessian matrix $\hat{H}_n^{\mathsf{dgm}} := \frac{1}{n}\left(X^{\mathsf{dgm}}\right)^\top X^{\mathsf{dgm}} - 4d_{\max}\sigma_{\varepsilon,\delta}^2 \cdot I$
4: $\hat{\mathbf{w}}_n^{\mathsf{dgm}} := \left(\hat{H}_n^{\mathsf{dgm}}\right)^{-1} \left(\frac{1}{n}\left(X^{\mathsf{dgm}}\right)^\top Y^{\mathsf{dgm}}\right)$.
5: **Return:** $\hat{\mathbf{w}}_n^{\mathsf{dgm}}$.

---

the Hessian matrix when training the model. However the de-bias operator introduces the possible inverse of a matrix with small eigenvalues, which severely hurts the performance of the learned model. We therefore propose a novel dataset release algorithm rather than the directly application to Gaussian mechanism – *Random Mixing prior to Gaussian Mechanism for Ordinary Least Squares (RMGM-OLS)*. The model learned from the corresponding released public dataset is also guaranteed to be asymptotically optimal, and, more importantly, avoids the problem of small eigenvalues.

## 4.1   DE-BIASED GAUSSIAN MECHANISM (DGM-OLS)

The De-biased Gaussian Mechanism for Ordinary Least Squares (DGM-OLS) includes the dataset release algorithm and the corresponding training algorithm. Algorithm 1 shows the overview and we will introduce them next.

**Dataset release algorithm.**   Each party directly applies Gaussian mechanism to their own dataset $D^j$ ($j = 1, \cdots m$) to satisfy the differential privacy. Consider two neighboring data matrices $D^j$ and $\left(D^j\right)'$ differing at exactly one row with the row index $i$. Implied by Assumption 2, we can compute the sensitivity of the data matrix $D^j$:

$$\left\| D^j - \left(D^j\right)' \right\| = \left\| D_i^j - \left(D_i^j\right)' \right\| \leq 2\sqrt{d_j} \leq 2\sqrt{d_{\max}}.$$

Then each party independently adds a Gaussian noise $R^j$ to $D^j$. Entries in $R^j$ are i.i.d sampled from Gaussian distribution $\mathcal{N}(0, 4d_{\max}\sigma_{\varepsilon,\delta}^2)$.

The dataset release algorithm meets the privacy constraints in section 3. No random matrix $B$ is shared among dif-

ferent parties. Lemma 1 guarantees that $\left(D^{\mathsf{dgm}}\right)^j$ is $(\varepsilon, \delta)$-differentially private w.r.t. $D^j$ for any $0 < \varepsilon \leq 1, \delta > 0$.

**Training algorithm.**   Given the dataset released through the above algorithm, there exists an asymptotic linear regression solution. Denote the feature matrix and the label vector of the private and public joint dataset as $[X, Y] = D$ and $[X^{\mathsf{dgm}}, Y^{\mathsf{dgm}}] = D^{\mathsf{dgm}}$. Further define $R := [R^1, \cdots, R^m] \in \mathbb{R}^{n \times (d+1)}$ and split $R$ into $R_X$ and $R_Y$ representing the additive noise to $X$ and $Y$ respectively.

Consider the ordinary least square solution for the public data $X^{\mathsf{dgm}}$ and $Y^{\mathsf{dgm}}$, whose explicit form is:

$$\left(\left(X^{\mathsf{dgm}}\right)^\top X^{\mathsf{dgm}}\right)\left(X^{\mathsf{dgm}}\right)^\top Y^{\mathsf{dgm}}. \tag{2}$$

Compared with our target solution $\mathbf{w}^* = \left(\mathbb{E}_{(\mathbf{x},y)\sim\mathcal{P}}\left[\mathbf{x}\mathbf{x}^\top\right]\right)^{-1}\mathbb{E}_{(\mathbf{x},y)\sim\mathcal{P}}[\mathbf{x}\cdot y]$, we can prove that $\mathrm{plim}_{n\to\infty}\frac{1}{n}\left(X^{\mathsf{dgm}}\right)^\top Y^{\mathsf{dgm}} = \mathbb{E}_{(\mathbf{x},y)\sim\mathcal{P}}[\mathbf{x}\cdot y]$ by the concentration of bounded random variables and multivariate normal distribution. Nevertheless, there is a gap between $\mathrm{plim}_{n\to\infty}\frac{1}{n}\left(X^{\mathsf{dgm}}\right)^\top X^{\mathsf{dgm}}$ and $\mathbb{E}_{(\mathbf{x},y)\sim\mathcal{P}}\left[\mathbf{x}\mathbf{x}^\top\right]$:

$$\mathrm{plim}_{n\to\infty} \frac{1}{n}\left(X^{\mathsf{dgm}}\right)^\top X^{\mathsf{dgm}}$$
$$= \mathrm{plim}_{n\to\infty} \frac{1}{n}\left(X^\top X + X^\top R_X + R_X^\top X + R_X^\top R_X\right)$$
$$= \mathbb{E}_{(\mathbf{x},y)\sim\mathcal{P}}\left[\mathbf{x}\mathbf{x}^\top\right] + 4d_{\max}\sigma_{\varepsilon,\delta}^2 \cdot I,$$

where the last equation again holds by the concentration of bounded random variables and multivariate normal distribution. To reduce the bias $4d_{\max}\sigma_{\varepsilon,\delta}^2 \cdot I$, we can revise the solution computation in Equation 2 to $\hat{\mathbf{w}}_n^{\mathsf{dgm}}$ defined as

$$\left(\frac{1}{n}\left(X^{\mathsf{dgm}}\right)^\top X^{\mathsf{dgm}} - 4d_{\max}\sigma_{\varepsilon,\delta}^2 \cdot I\right)^{-1} \left(\frac{1}{n}\left(X^{\mathsf{dgm}}\right)^\top Y^{\mathsf{dgm}}\right).$$

The first term is estimated for the inverse of the Hessian matrix $\mathbb{E}_{(\mathbf{x},y)\sim\mathcal{P}}\left[\mathbf{x}\mathbf{x}^\top\right]$, which we denote as $(\hat{H}_n^{\mathsf{dgm}})^{-1}$. The asymptotic optimality for the solution $\hat{\mathbf{w}}_n^{\mathsf{dgm}}$ is implied by the theorem below and the proof is in the Appendix.

**Theorem 1.** *When $\beta \leq c$ for some variable $c$ that is dependent of $\sigma_{\varepsilon,\delta}$, $d$, and $\mathcal{P}$, but is independent of $n$,*

$$\mathbb{P}\left[\|\hat{\mathbf{w}}_n^{\mathsf{dgm}} - \mathbf{w}^*\| > \beta\right] < \exp\left(-\tilde{O}\left(\beta^2 \frac{n}{\sigma_{\varepsilon,\delta}^4 d^2 d_{\max}^2}\right)\right),$$

**Problem of small eigenvalues.**   The expectation of $\hat{H}_n^{\mathsf{dgm}}$ is a positive definite matrix given Assumption 3, but the sample of $\hat{H}_n^{\mathsf{dgm}}$ itself is not guaranteed. With a certain probability, it has small eigenvalues that might lead to explosion when computing its inverse. In our experiments (section 5), we find that $\hat{H}_n^{\mathsf{dgm}}$ suffers from the small eigenvalues even if $n$ is as large as $10^6$. As a result, the model utility is much more inferior than what is guaranteed theoretically. This motivates us to design the second algorithm.

---
**Algorithm 2** RMGM-OLS

**Dataset Release**

1: **Input:** $D = \left[D^1, \cdots, D^m\right], \varepsilon, \delta, k$.
2: The first party pre-generates a $k \times n$ random matrix $B$ where all entries in $B$ are *i.i.d.* sampled from the distribution with probability $1/2$ for 1 and $1/2$ for $-1$. Then first party sends the random matrix sample $B$ to all parties.
3: **for** $j = 1, \cdots, m$ **do**
4:     The party $j$ computes $(D^{\mathsf{rmgm}})^j := BD^j/\sqrt{k} + R^j$, where $R^j$ is a $k \times d_j$ random matrix and all elements in $R^j$ are *i.i.d.* sampled from the multivariate normal distribution $\mathcal{N}\left(0, 4d_{\max}\sigma_{\varepsilon,\delta}^2\right)$.
5: **end for**
6: **Return:** $D^{\mathsf{rmgm}} := \left[(D^{\mathsf{rmgm}})^1, \cdots, (D^{\mathsf{rmgm}})^m\right]$.

**Training Algorithm**

1: **Input:** $D^{\mathsf{rmgm}}, \varepsilon, \delta$
2: $[X^{\mathsf{rmgm}}, Y^{\mathsf{rmgm}}] = D^{\mathsf{rmgm}}$
3: Compute the ordinary least square solution
$\hat{\mathbf{w}}_n^{\mathsf{rmgm}} := \left((X^{\mathsf{rmgm}})^\top X^{\mathsf{rmgm}}\right)^{-1}(X^{\mathsf{rmgm}})^\top Y^{\mathsf{rmgm}}$.
4: **Return:** $\hat{\mathbf{w}}_n^{\mathsf{rmgm}}$.

---

## 4.2 RANDOM MIXING PRIOR TO GAUSSIAN MECHANISM (RMGM-OLS)

In previous method's dataset release stage, when we directly add the Gaussian additive noise $R$ to the data, in order to guarantee DP, the norm of the noise needed has to be the same order (in $n$) as the norm of the data matrix $D$. Both $D$ and $R$ have norm in $\Theta(\sqrt{n})$. Thus later in the training stage, the additive noise $R$ when compared to the data matrix $X$ would not diminish as $n \to \infty$ and we have to subtract $4d_{\max}\sigma_{\varepsilon,\delta}^2 \cdot I$ from $\left(X^{\mathsf{dgm}}\right)^\top X^{\mathsf{dgm}}$ to remove this additive noise in order to obtain the optimal model weights. This subtraction is the problematic part that brings training instability (small eigenvalues in the Hessian matrix).

Instead, we can avoid such subtraction in the training stage by imposing a smaller noise in the data release stage. If we can design the data release stage properly, so that the addictive noise has relatively smaller order in $n$ than $D$, in the later training stage, the learner would no longer need the problematic de-biasing step.

Algorithm 2 shows the full details of Random Mixing prior to Gaussian Mechanism for Ordinary Least Squares (RMGM-OLS). We now explain the design of data release and training algorithm based on the above insights.

**Dataset release algorithm.** Suppose $\mathbf{b}$ is an $n$-dimensional vector in $\{-1, 1\}^n$. For any two neighbouring daasets $D^j$ and $\left(D^j\right)'$ that are different at row index $i$, the sensitivity of $\mathbf{b}^\top D^j$ is

$$\left\|\mathbf{b}^\top D^j - \mathbf{b}^\top \left(D^j\right)'\right\| = \left\|D_i^j - \left(D_i^j\right)'\right\| \le 2\sqrt{d_j} \le 2\sqrt{d_{\max}}.$$

Moreover, when $B \in \{-1, 1\}^{k \times n}$, $BD^j/\sqrt{k}$ has sensitivity $2\sqrt{d_{\max}}$ as well.

We now introduce the data release algorithm. Suppose all parties are sharing a random matrix $B \in \{-1, 1\}^{k \times n}$, where all elements in $B$ are *i.i.d.* sampled from the distribution with probability $1/2$ for 1 and $1/2$ for $-1$. Then we define the local computation for each party $j$:

$$(D^{\mathsf{rmgm}})^j := BD^j/\sqrt{k} + R^j,$$

where $R^j$ is a $k \times d_j$ random matrix and all elements in $R^j$ are *i.i.d.* sampled from the multivariate normal distribution $\mathcal{N}\left(0, 4d_{\max}\sigma_{\varepsilon,\delta}^2\right)$. Gaussian mechanism guarantees for any fixed $B \in \{1, -1\}^{k \times n}$, $(D^{\mathsf{rmgm}})^j$ is $(\varepsilon, \delta)$-differentially private *w.r.t.* the dataset $D^j$ for $0 < \varepsilon \le 1, \delta > 0$.

Importantly, now the addictive noise $R^j$ is relatively small than $BD^j/\sqrt{k}$. The order of $\|R^j\|$ is $\Theta(k)$ while the order of $\left\|BD^j/\sqrt{k}\right\| \approx \|D^j\|$ is $\Theta(n)$ (by JL Lemma). If we set $k = o(n)$, the additive noise compared to the original data matrix $D$ will diminish as $n \to \infty$. This implies that the standard ordinary least square solution to the public dataset $[X^{\mathsf{rmgm}}, Y^{\mathsf{rmgm}}]$ would converge to the optimal solution $\mathbf{w}^*$ without special subtraction.

**Training algorithm.** Given the feature matrix $X^{\mathsf{rmgm}}$ and the label vector $Y^{\mathsf{rmgm}}$ from the released dataset, we show that the vanilla ordinary least square solution

$$\hat{\mathbf{w}}_n^{\mathsf{rmgm}} := \left((X^{\mathsf{rmgm}})^\top X^{\mathsf{rmgm}}\right)^{-1}(X^{\mathsf{rmgm}})^\top Y^{\mathsf{rmgm}}$$

is asymptotically optimal, i.e. $\mathrm{plim}_{n \to \infty} \hat{\mathbf{w}}_n^{\mathsf{rmgm}} = \mathbf{w}^*$.

To prove the above asymptotic optimality, we show $\mathrm{plim}_{n \to \infty}(X^{\mathsf{rmgm}})^\top X^{\mathsf{rmgm}} = \mathbb{E}_{(\mathbf{x},y)\sim\mathcal{P}}\left[\mathbf{x}\mathbf{x}^\top\right]$ and $\mathrm{plim}_{n \to \infty}(X^{\mathsf{rmgm}})^\top Y^{\mathsf{rmgm}} = \mathbb{E}_{(\mathbf{x},y)\sim\mathcal{P}}[\mathbf{x} \cdot y]$ respectively, and together they prove the optimality.

Define $R = \left[R^1, \cdots, R^m\right] \in \mathbb{R}^{k \times (d+1)}$ and split $R$ into $R_X$ and $R_Y$ representing the additive noises to $BX/\sqrt{k}$ and $BY/\sqrt{k}$ respectively. Because $\mathrm{plim}_{n \to \infty} \frac{1}{n}X^\top X = \mathbb{E}_{(\mathbf{x},y)\sim\mathcal{P}}\left[\mathbf{x}\mathbf{x}^\top\right]$, it is sufficient to show $\mathrm{plim}_{n \to \infty} \frac{1}{n}(X^{\mathsf{rmgm}})^\top X^{\mathsf{rmgm}} - \frac{1}{n}X^\top X = \mathbf{0}$. Now we decompose $\frac{1}{n}(X^{\mathsf{rmgm}})^\top X^{\mathsf{rmgm}} - \frac{1}{n}X^\top X$ as below:

$$\underbrace{\frac{1}{n}\left(X^\top \frac{B^\top B}{k} X - X^\top X\right)}_{\text{Lemma 2}} + \underbrace{\frac{1}{n}\left(X^\top \frac{B^\top}{\sqrt{k}} R_X + R_X^\top \frac{B}{\sqrt{k}} X\right)}_{\text{cvg. of gauss. dist.}} + \underbrace{\frac{1}{n}R_X^\top R_X}_{k = o(n)}.$$

We informally show how each term converges to $\mathbf{0}$ as $n \to \infty$:

1. $\frac{1}{n}\left(X^\top \frac{B^\top B}{k} X - X^\top X\right)$. If $k \to \infty$ as $n \to \infty$, the convergence is directly implied by Lemma 2.

2. $\frac{1}{n}\left(X^\top \frac{B^\top}{\sqrt{k}}R_X + R_X^\top \frac{B}{\sqrt{k}}X\right)$. Properties of normal distribution guarantees the approximation $\frac{B^\top}{\sqrt{k}}R_X \approx R'_X$, where $R'_X \in \mathbb{R}^{n\times d}$ is a Gaussian matrix with $\mathcal{N}\left(0, 4d_{\max}\sigma_{\varepsilon,\delta}^2\right)$. Then $\left\|\frac{1}{n}X^\top \frac{B^\top}{\sqrt{k}}R_X\right\| \approx \left\|\frac{1}{n}X^\top R'_X\right\| = O\left(\frac{1}{\sqrt{n}}\right)$.

3. $\frac{1}{n}R_X^\top R_X$. If $k \to \infty$ as $n \to \infty$, $\left\|\frac{1}{k}R_X^\top R_X\right\|$ will converge to $4d_{\max}\sigma_{\varepsilon,\delta}^2 \cdot I$. On the other hand, when $k = o(n)$, $\frac{k}{n}\cdot 4d_{\max}\sigma_{\varepsilon,\delta}^2 \cdot I$ will converge to $\mathbf{0}$ as $n \to \infty$.

Notice that the above convergence relies on the proper selection of $k$. There exists a trade-off: larger $k$ leads to better convergence rate of the first term, but worse rate for the diminishing of additive noise – the third term. The following theorem shows the exact asymptotic rate:

**Theorem 2.** *When $\beta \le c$ for some variable $c$ that is dependent of $d$ and $\mathcal{P}$, but independent of $\sigma_{\varepsilon,\delta}$, $n$, we have*

$$\mathbb{P}\left[\|\hat{\mathbf{w}}_n^{\mathsf{rmgm}} - \mathbf{w}^*\| > \beta\right] <$$
$$\exp\left(-O\left(\min\left\{\frac{k\beta^2}{d^2}, \frac{n\beta}{kdd_{\max}\sigma_{\varepsilon,\delta}^2}, \frac{n^{1/2}\beta}{dd_{\max}^{1/2}\sigma_{\varepsilon,\delta}}\right\}\right) + \tilde{O}(1)\right).$$

*If we choose $k = O\left(\frac{n^{1/2}d^{1/2}}{d_{\max}^{1/2}\sigma_{\varepsilon,\delta}}\right)$, then*

$$\mathbb{P}\left[\|\hat{\mathbf{w}}_n^{\mathsf{rmgm}} - \mathbf{w}^*\| > \beta\right] <$$
$$\exp\left(-\frac{n^{1/2}\beta}{d^{3/2}d_{\max}^{1/2}\sigma_{\varepsilon,\delta}} \cdot O\left(\min\{1, \beta\}\right) + \tilde{O}(1)\right).$$

In the theorem, $k$ is selected to balance $\frac{k\beta^2}{d^2}$ and $\frac{n\beta}{kdd_{\max}\sigma_{\varepsilon,\delta}^2}$. To achieve the optimal rate for $f(\beta)$ with any fixed $\beta$, the optimal $k$ is chosen as $O\left(\frac{n^{1/2}d^{1/2}}{d_{\max}^{1/2}\sigma_{\varepsilon,\delta}}\right)$.

**Comparison with DGM-OLS.** The near-zero eigenvalue issue is solved since $(X^{\mathsf{rmgm}})^\top X^{\mathsf{rmgm}} \succeq \mathbf{0}$ holds naturally by its definition. Moreover, although the convergence rate of $n$ is sacrificed, the orders in $d, d_{\max}$ and $\sigma_{\varepsilon,\delta}$ are much improved. In section 5 we show that the RMGM-OLS outperforms DGM-OLS on both synthetic datasets even when $n$ is as large as $3 \times 10^6$.

# 5 EXPERIMENTAL EVALUATION

In this section, we evaluate DGM-OLS and RMGM-OLS on both synthetic and real world datasets. Our experiments on synthetic dataset are designed to verify the theoretical asymptotic results in section 4 by increasing the training set size $n$. We further justify the algorithm performance on five real-world datasets, four from UCI Machine Learning Repository[1] [Dua and Graff, 2017] and one from kaggle.

---

[1] https://archive-beta.ics.uci.edu/ml/datasets

## 5.1 EXPERIMENT SET-UP

**Algorithm set-up.** We evaluate both DGM-OLS and RMGM-OLS. For $k$ in RMGM-OLS, we set $k = \frac{\sqrt{n}}{\sigma_{\varepsilon,\delta}}$ in synthetic dataset experiments and select the best $k$ from $\{10^2, 3\times10^2, 10^3, 3\times10^3, 10^4\}$ in real-world dataset experiments. Because of the numerical instability of computing Hessian inverse mentioned early, we add small $\lambda \cdot I$ with $\lambda = 10^{-5}$ to all Hessian matrices.

**Baseline.** In addition, we consider the following baselines to help qualify the performance of proposed algorithms.

- *OLS*: The explicit solution for linear regression given training data $(X, Y)$ and serves as the performance's upper bound for private algorithms, i.e. non-private solution.
- *Biased Gaussian mechanism (BGM-OLS)*: The same data release algorithm in DGM-OLS, but has a different training algorithm. Given a released dataset $(X^{\mathsf{bgm}}, Y^{\mathsf{bgm}})$ by Gaussian mechanism, BGM-OLS outputs the vanilla ordinary least square solution $\hat{\mathbf{w}}_n^{\mathsf{bgm}} = \left(\left(X^{\mathsf{bgm}}\right)^\top X^{\mathsf{bgm}}\right)^{-1}\left(X^{\mathsf{bgm}}\right)^\top Y^{\mathsf{bgm}}$. In other words, it is DGM-OLS without training debiasing.

**Evaluation metric.** In the experiments on synthetic datasets, we estimate the probability of the $\ell^2$ distance between the model weights $\hat{\mathbf{w}}_n$ from each algorithm or baseline and the ground truth model weight $\mathbf{w}^*$:

$$\mathbb{P}\left(\|\hat{\mathbf{w}}_n - \mathbf{w}^*\| > \beta\right).$$

We also evaluate the expectation of the $\ell^2$ distance between weights for different algorithms:

$$\mathbb{E}\left\|\hat{\mathbf{w}}_n - \mathbf{w}^*\right\|.$$

If an algorithm is asymptotically optimal, we can see both $\mathbb{P}\left(\|\hat{\mathbf{w}}_n - \mathbf{w}^*\| > \beta\right)$ and $\mathbb{E}\left\|\hat{\mathbf{w}}_n - \mathbf{w}^*\right\|$ converge to 0 when $n$ increases.

For the experiments on real world datasets, we evaluate learned models $\hat{\mathbf{w}}_n$ by the mean squared loss on the test set.

## 5.2 EVALUATION ON SYNTHETIC DATASETS

**Data generation.** We define the feature dimension $d = 10$. Each weight value of the ground truth linear model $\mathbf{w}^*$ is independently sampled from uniform distribution between $-1/d$ and $1/d$. A single data point $(\mathbf{x}, y)$ is sampled as the following: each feature value in $\mathbf{x}$ is independently sampled from a uniform distribution between $-1$ and $1$; label $y$ is computed as $(\mathbf{w}^*)^\top \mathbf{x}$. Two assumptions for the data distribution $\mathcal{P}$, Assumption 2 and Assumption 3, can be verified. Moreover, we set 6 parties in total, 5 of which have 2 attributes and the remaining one has 1 attribute.

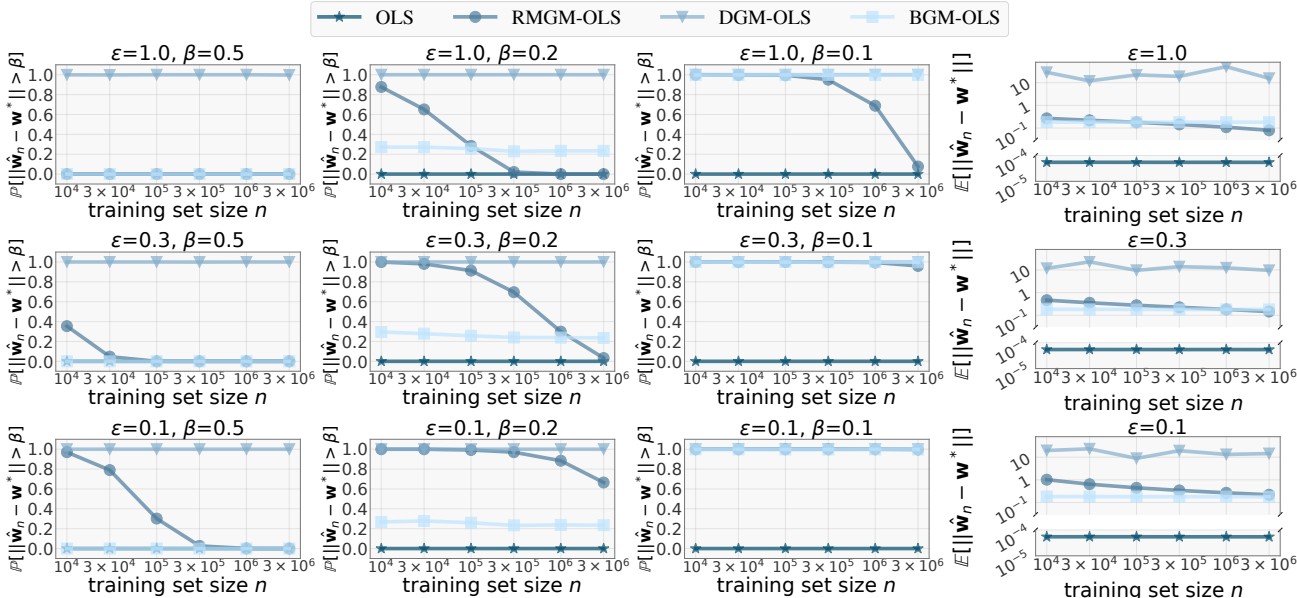

Figure 2: $\mathbb{P}\left[\|\hat{\mathbf{w}}_n - \mathbf{w}^*\| > \beta\right]$ and $\mathbb{E}\left[\|\hat{\mathbf{w}}_n - \mathbf{w}^*\|\right]$ as dataset size $n$ increases for different algorithms when $\varepsilon = 1.0, 0.3, 0.1$. For all pairs of $(\varepsilon, \beta)$ except two most extreme cases $(0.3, 0.1)$ and $(0.1, 0.1)$, RMGM-OLS shows asymptotic tendencies $\text{plim}_{n\to\infty}\mathbb{P}\left[\|\hat{\mathbf{w}}_n^{\text{rmgm}} - \mathbf{w}^*\| > \beta\right] = 0$. DGM-OLS does not show such tendencies even when training set size $n$ is as large as $3 \times 10^6$.

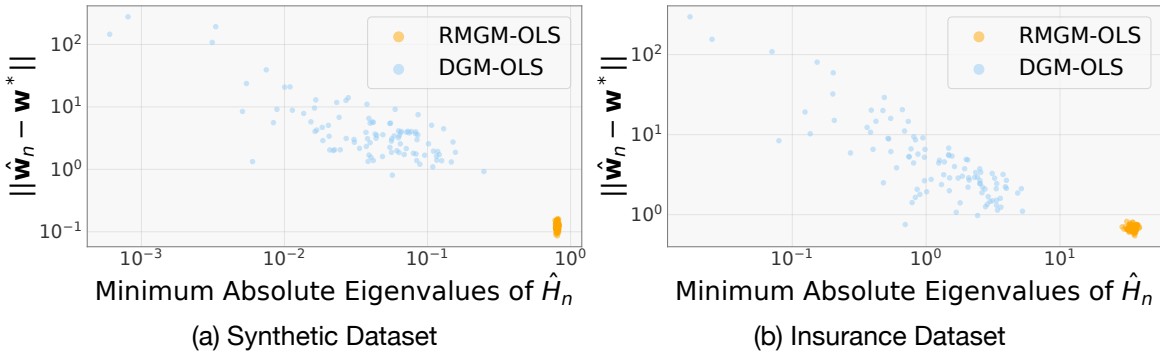

Figure 3: Scatter plots of $\ell_2$ distance versus minimum absolute eigenvalue of Hessian matrix. The left figure is for the synthetic dataset when $n = 10^6$ and $\varepsilon = 1.0$. The right figure is for the *Insurance* dataset when $\varepsilon = 1.0$. Each point is processed by a different random seed for DGM-OLS and RMGM-OLS. Both figures show that the Hessian matrix in DGM-OLS is more likely to have small eigenvalues, which further lead to large distance $\|\hat{\mathbf{w}}_n - \mathbf{w}^*\|_2$.

**Results.** We vary the training set size $n \in \{10^4, 3 \times 10^4, 10^5, 3 \times 10^5, 10^6, 3 \times 10^6\}$ and privacy budget $\varepsilon \in \{1, 0.3, 0.1\}$ with fixed $\delta = 10^{-5}$. We estimate the $\mathbb{P}\left[\|\mathbf{w}_n - \mathbf{w}^*\| > \beta\right]$ and $\mathbb{E}\|\mathbf{w}_n - \mathbf{w}^*\|$ for different algorithms with 1000 random seeds. Figure 2 shows how $\mathbb{P}\left[\|\mathbf{w}_n - \mathbf{w}^*\| > \beta\right]$ and $\mathbb{E}\|\mathbf{w}_n - \mathbf{w}^*\|$ of each algorithm change when training set size $n$ increases.

Regarding two baselines, $\mathbb{P}\left[\|\mathbf{w}_n - \mathbf{w}^*\| > \beta\right]$ of OLS solutions, without any private constraint, are close to the ground truth $\mathbf{w}^*$ under all $\beta$ with probability 0. Nonetheless, $\mathbb{P}\left[\|\mathbf{w}_n - \mathbf{w}^*\| > \beta\right]$ of BGM-OLS keeps mostly unchanged as $n$ increases. Especially, $\mathbb{P}\left[\|\mathbf{w}_n - \mathbf{w}^*\| > 0.1\right]$

stays at 1 for all $n$. Such results are expected in BGM-OLS's convergence: $\text{plim}_{n\to\infty} \frac{1}{n}\left(X^{\text{bgm}}\right)^\top X^{\text{bgm}} = \mathbb{E}_{\mathbf{x}}\left[\mathbf{x}\mathbf{x}^\top\right] + 4d_{\max}\sigma_{\varepsilon,\delta}^2 \cdot I$, which introduces a non-diminishing bias $4d_{\max}\sigma_{\varepsilon,\delta}^2 \cdot I$.

Next, we compare DGM-OLS and RMGM-OLS. RMGM-OLS outperforms DGM-OLS at both the convergence of probability $\mathbb{P}\left[\|\mathbf{w}_n - \mathbf{w}^*\| > \beta\right]$ (the first three figures in Figure 2) and the expected distance $\mathbb{E}\left[\|\mathbf{w}_n - \mathbf{w}^*\|\right]$ (the last figure in Figure 2). RMGM-OLS shows the asymptotic tendencies in all values of $\beta$ when $\varepsilon = 1.0$. Although DGM-OLS has better rate at $n$ than RMGM-OLS theoretically, $n = 3 \times 10^6$ is not large enough to show the asymptotic

tendencies for DGM-OLS.

DGM-OLS is even much worse than BGM-OLS, which is almost random guess. It is caused by the small eigenvalue issue discussed in section 4. To illustrate it, Figure 3 (a) shows the scatter plot, where the $x$-axis is minimum eigenvalues of the Hessian matrix $\hat{H}_n$ and $y$-axis is the distance between our solutions and the optimal solution $\|\hat{\mathbf{w}}_n - \mathbf{w}^*\|$. Each point is processed by a different random seed for DGM-OLS and BGM-OLS when $n = 10^6$ and $\varepsilon = 1.0$. $\|\hat{\mathbf{w}}_n - \mathbf{w}^*\|$ and the minimum absolute eigenvalues of $\hat{H}_n$ have a strong positive correlation. With a certain probability, the minimum eigenvalue of DGM-OLS is smaller than $10^{-2}$ and corresponding $\|\mathbf{w}_n - \mathbf{w}^*\|$ is larger than 10.

Overall RMGM-OLS has the best empirical performance across various settings of $\varepsilon$ and $n$ on the synthetic data, as its asymptotically optimality is verified and it consistently outperforms two other private algorithms when $n$ is large enough. Though DGM-OLS seems to have stronger theoretical guarantee in the aspect of rate in $n$, its poor empirical performance comes from two aspects: 1. small eigenvalues occur due to the design of the training algorithm; 2. extremely large $n$ is necessary to show the asymptotic optimality due to the worse rates of $d, d_{\max}$ and $\sigma_{\varepsilon,\delta}$.

## 5.3 EVALUATION ON REAL WORLD DATASETS

**Dataset.** We experiment with five datasets:

- *Insurance* [Lantz, 2019]: predicting the insurance premium from features including age, bmi, expenses, etc.
- *Bike* [Fanaee-T and Gama, 2014]: predicting the count of rental bikes from features such as season, holiday, etc.
- *Superconductor* [Hamidieh, 2018]: predicting critical temperature from chemical features.
- *GPU* [Ballester-Ripoll et al., 2019, Nugteren and Codreanu, 2015]: predicting Running time for multiplying two $2048 \times 2048$. matrices using a GPU OpenCL SGEMM kernel with varying parameters.
- *Music Song* [Bertin-Mahieux et al., 2011]: predicting the release year of a song from audio features.

We split the original dataset into train and test by the ratio $4 : 1$. The number of training data $n$, the number of features $d$ and the number of parties are listed in Table 1. The attributes are evenly distributed among parties. All features and labels are normalized into $[0, 1]$.

**Results.** For each dataset, we evaluate OLS and three differentially private algorithms by the mean squared loss on the test split. Table 1 shows the results for $\varepsilon \in \{0.1, 0.3, 1.0\}$ and $\delta = 10^{-5}$. We can check that the loss of DGM-OLS is usually much larger than others and RMGM-OLS achieves the lowest losses for most cases (12 out of 15). Moreover, Figure 3 (b) shows that DGM-OLS has the small eigenvalue problem as well in the real world dataset

experiments. These results are consistent with the results on synthetic dataset. We therefore recommend RMGM-OLS as a practical solution to privately release the dataset and build the linear regression models.

## 6 RELATED WORK

**Differentially private dataset release.** Many recent works [Sheffet, 2017, Gondara and Wang, 2020, Xie et al., 2018, Jordon et al., 2018, Lee et al., 2019, Xu et al., 2017, Kenthapadi et al., 2012] study the differentially private data release algorithms. However, those algorithms either only serve for data release from a *single-party* [Sheffet, 2017, Gondara and Wang, 2020], or focus on the feature dimension reduction or empirical improvement [Lee et al., 2019, Xu et al., 2017, Kenthapadi et al., 2012], which is orthogonal to the study of asymptotical optimality w.r.t. dataset size. In Sheffet [2017] and Gondara and Wang [2020], the random Gaussian projection matrices in their method contribute to the differential privacy guarantee, hence the sharing of projection matrix would violate the privacy guarantee between parties. Nevertheless, without sharing this projection matrix, the utility cannot be guaranteed anymore. In Xie et al. [2018] and Jordon et al. [2018], they train a differentially private GAN. However, it is not obvious to rigorously privately share data information during their training when each party holds different attributes but same instances. Lee et al. [2019] proposes a random mixing method and also analyzes the linear model. However, the way they mix only works for realizable linear data. It is not able to be extended to the general linear regression and the asymptotic optimality guarantee. Xu et al. [2017] and Kenthapadi et al. [2012] focus on the feature dimension reduction, which is orthogonal to the study of asymptotical optimality w.r.t. dataset size.

**Asymptotically optimal differentially private convex optimization.** A large amount of work study differentially private optimization for convex problems [Bassily et al., 2014, 2019, Feldman et al., 2020] or particularly for linear regression [Sheffet, 2017, Kasiviswanathan et al., 2011, Chaudhuri and Hsu, 2012]. They mainly differ from our work in the sense that their goal is to release the final model while ours is to release the dataset.

**Linear regression in vertical federated learning.** Linear regression is a fundamental machine learning task. Hall et al. [2011], Nikolaenko et al. [2013], Gascón et al. [2017] studying linear regression over vertically partitioned datasets based on secure multi-party computation. However, cryptographic protocols such as Homomorphic Encryption [Hall et al., 2011, Nikolaenko et al., 2013] and garbled circuits [Nikolaenko et al., 2013, Gascón et al., 2017] lead to heavy overhead on computation and communication. From this aspect, DP-based techniques are more practical.

| Dataset | Statistics | Method | | | | | | | | | |
|---|---|---|---|---|---|---|---|---|---|---|---|
| | | OLS | $\varepsilon = 1.0$ | | | $\varepsilon = 0.3$ | | | $\varepsilon = 0.1$ | | |
| | | | DGM | RMGM | BGM | DGM | RMGM | BGM | DGM | RMGM | BGM |
| Insurance | $n = 1070$, $d = 9$, $m = 5$ | 0.008 | 0.7015 | **0.0791** | 0.0805 | 0.7550 | **0.0782** | 0.0850 | 0.7263 | **0.0793** | 0.0832 |
| Bike | $n = 13903$, $d = 13$, $m = 5$ | 0.017 | 0.8105 | **0.0581** | 0.0691 | 0.9080 | 0.0711 | **0.0703** | 0.8792 | **0.0700** | 0.0707 |
| Superconductor | $n = 17010$, $d = 81$, $m = 10$ | 0.009 | 0.9794 | **0.0659** | 0.0670 | 1.0075 | 0.0707 | **0.0704** | 0.9220 | **0.0699** | 0.0704 |
| GPU | $n = 193280$, $d = 14$, $m = 5$ | 0.007 | 0.6953 | **0.0137** | 0.0158 | 0.7843 | **0.0160** | **0.0160** | 0.7822 | 0.0165 | **0.0160** |
| Music Song | $n = 412276$, $d = 90$, $m = 10$ | 0.011 | 1.0167 | **0.0202** | 0.7194 | 1.6462 | **0.1039** | 0.7479 | 1.5583 | **0.5654** | 0.7508 |

Table 1: Mean squared losses on real world datasets. RMGM-OLS achieves the lowest losses in most settings (12 out of 15).

# 7 CONCLUSION

We propose and analyze two differentially private algorithms under multi-party setting for linear regression, and theoretically both of them are asymptotically optimal with increasing dataset size. Empirically, RMGM-OLS has the best performance on both synthetic datasets and real-world datasets, while extremely large training set size $n$ is necessary for DGM-OLS. We hope our work can bring more attention to the need for multi-party data release algorithms and we believe that ML practitioners would benefit from such effort in the era of privacy.

**Future work.** We focus on linear regression only, and one future direction is to extend our algorithm to classification, e.g. logistic regression, while achieving the same asymptotic optimality. In addition, we assume different parties own the same set of data subjects. Another future direction is to relax this assumption: the set of subjects owned by different parties might be slightly different.

### Acknowledgements

The authors thank Xiaowei Zhang for drawing pictures in Figure 1. RW and KQW are supported by grants from the National Science Foundation NSF (IIS-2107161, III-1526012, IIS-1149882, and IIS-1724282), and the Cornell Center for Materials Research with funding from the NSF MRSEC program (DMR-1719875), and SAP America.

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
