# OpenReview forum: "Differentially Private Multi-Party Data Release for Linear Regression"
_auai.org/UAI/2022/Conference — UAI 2022 Poster_

### Official Review · Reviewer_73U7 · 2022-04-10

**Q2(1) Originality/Novelty:** 3
**Q2(2) Significance/Impact:** 3
**Q2(3) Correctness/Technical Quality:** 2
**Q2(6) Clarity Of Writing:** 3
**Q6 Overall Score:** 6
**Q8 Confidence In Your Score:** 3

**Q1 Summary And Contributions:**

This paper focuses on safe data release in a multi-party scenario, where different stakeholders own disjoint sets of attributes belonging to the same group of data subjects. Two differential private algorithms for linear regression in multi-party scenarios are proposed and theoretically proved to be asymptotically optimal with increasing dataset size.

**Q10 Ethical Concerns (Optional):**

No.

**Q2 Assessment Of The Paper:**

More detailed information regarding each of these aspects is given below:

**Q2(4) Quality Of Experiments (Optional):**

3: Good: The experimental evaluation is adequate, and the results convincingly support the main claims.

**Q2(5) Reproducibility:**

3: Good: Key resources (e.g., proofs, code, data) are available and key details (e.g., proofs, experimental setup) are sufficiently well-described for competent researchers to confidently reproduce the main results.

**Q3 Main Strengths:**

1. The paper proposes a local differential privacy framework to deal with the problem of Isolated Data Island, which is interesting.
3. The paper provides a good proof that two algorithms are asymptotically optimal.


**Q4 Main Weakness:**

1. The discussion of security in this paper is not sufficient. For example, if the shared matrix is leaked, will it cause adverse effects?
2. The concerns in this paper are very similar to the federated learning scenario, why not consider federated learning as the baseline for comparison?


**Q5 Detailed Comments To The Authors:**

The basic idea of this work is attractive, but there is still room for improvement. Questions and comments are as follows:
1. Will the shared matrix have adversely affect if it is leaked?
2. Different organizations have different attributes of the same subject. How do you combine the attributes of the same subject when transferring data in reality? Is there a risk of leakage in the alignment or combining process?
3. Federated learning also has similar scenarios. The paper does not give sufficient reasons for not using federated learning.



**Q7 Justification For Your Score:**

This paper presents a new LDP idea to deal with data the problem of Isolated Data Island. Although experimental analysis and comparison are not sufficient, it is still an interesting work.

**Q9 Complying With Reviewing Instructions:**

1: Yes.

---

### Official Review · Reviewer_mg9s · 2022-04-13

**Q2(1) Originality/Novelty:** 2
**Q2(2) Significance/Impact:** 3
**Q2(3) Correctness/Technical Quality:** 3
**Q2(6) Clarity Of Writing:** 3
**Q6 Overall Score:** 7
**Q8 Confidence In Your Score:** 3

**Q1 Summary And Contributions:**

The paper considers differentially private release of vertically partitioned data and linear regression training with it. Two algorithms are considered (both theoretically and experimentally): plain Gaussian mechanism where Gaussian noise is added to the data, and a random projection based mechanism which reduces the dimensionality (and the amount of noise added thereby).

**Q2 Assessment Of The Paper:**

More detailed information regarding each of these aspects is given below:

**Q2(4) Quality Of Experiments (Optional):**

3: Good: The experimental evaluation is adequate, and the results convincingly support the main claims.

**Q2(5) Reproducibility:**

3: Good: Key resources (e.g., proofs, code, data) are available and key details (e.g., proofs, experimental setup) are sufficiently well-described for competent researchers to confidently reproduce the main results.

**Q3 Main Strengths:**

Very well written paper, which clearly shows the strength of the random projection (and JL transform) based approach compared to the plain Gaussian mechanism. Both theory and experimental sections seem strong.

**Q4 Main Weakness:**

I found a bit confusing that there is 'multiparty setting' mentioned several times, this made me to expect there would be something cryptography related mentioned. Also, 'vertically partitioned data' is so widely used term for this situation that I think it might be good to address that more explicitly even if you do not decide to use that term.

I think there would be more to discuss about the saving brought by the random projection approach. For example, since you reduce the dimension, the communication costs will be smaller and if you hade some multiparty computation protocol, for example, having fewer entries would mean less computation.

**Q5 Detailed Comments To The Authors:**

The issues related to communication savings and computation savings in case of MPC (or some related protocol) brought by the random projections might be good to discuss, and I believe one can find some literature related to that as well.

**Q7 Justification For Your Score:**

Both theory and experimental sections seem strong, and I think the random projection aspect of DP has not been addressed that widely in privacy-preserving ML literature so I think this would be a good fit for this venue.

**Q9 Complying With Reviewing Instructions:**

1: Yes.

---

### Official Review · Reviewer_fTaR · 2022-04-17

**Q2(1) Originality/Novelty:** 2
**Q2(2) Significance/Impact:** 3
**Q2(3) Correctness/Technical Quality:** 3
**Q2(6) Clarity Of Writing:** 3
**Q6 Overall Score:** 5
**Q8 Confidence In Your Score:** 2

**Q1 Summary And Contributions:**

This work proposes a new method for **multi-party** Privacy-preserving data release for linear regression. Existing methods mostly focus on single-party settings. Naive methods to solve the multi-party problem are either unrealistic or have lower utility. The proposed DGM-OLS and RMGM-OLS, it is guaranteed to converge to optimal solutions asymptotically. Experiments on synthetic and real-world datasets show RMGM-OLS works well.


**Q2 Assessment Of The Paper:**

More detailed information regarding each of these aspects is given below:

**Q2(4) Quality Of Experiments (Optional):**

2: Fair: The experimental evaluation is weak: important baselines are missing, or the results do not adequately support the main claims.

**Q2(5) Reproducibility:**

2: Fair: Key resources (e.g., proofs, code, data) are unavailable but key details (e.g., proof sketches, experimental setup) are sufficiently well-described for an expert to confidently reproduce the main results.

**Q3 Main Strengths:**

1. This paper works on an important problem -- privacy under multi-party setting, which was underexplored.
2. They point out the issue of small eigenvalues of the straightforward solution DGM-OLS.
3. The proposed RMGM-OLS outperforms the baselines and DGM-OLS in most cases.

**Q4 Main Weakness:**

1. The proposed methods can only be applied to linear regression with OLS.
2. It is better to avoid claiming DGM-OLS as a proposed method as it does not really work.


**Q5 Detailed Comments To The Authors:**

1. Figure 2: It seems the colors of legend does not exactly match those in the figure.
2. Is there any results (similar to Figure 2) for different values of epision?
3. It is better to also show the eigenvalues comparison for the real-world datasets if it is possible. This will help explain why DGM-OLS does not work practically although it is theoretically guaranteed to achieve optimal results with infinite sample.

**Q7 Justification For Your Score:**

1. The paper is well presented.
2. Theoretical results are solid to explain why the proposed methods may work and why DGM-OLS can fail in practice.
3. Experiments are satisfactory.

**Q9 Complying With Reviewing Instructions:**

1: Yes.

---

### Official Review · Reviewer_jHG1 · 2022-04-17

**Q2(1) Originality/Novelty:** 2
**Q2(2) Significance/Impact:** 2
**Q2(3) Correctness/Technical Quality:** 3
**Q2(6) Clarity Of Writing:** 3
**Q6 Overall Score:** 5
**Q8 Confidence In Your Score:** 3

**Q1 Summary And Contributions:**

This paper studies differential privacy (DP) in the setting of multiple independently released datasets for regression. Two DP algorithms , RMGM-OLS and DGM-OLS are proposed. Theoretical analysis is presented on the DP guarantee w.r.t. the Gaussian mechanism. Experiments with synthetic data demonstrates that RMGM-OLS converges to the optimal (non-private) solutions with the increase of dataset size, and experiments with real data show the superior utility achieved by RMGM-OLS.


**Q2 Assessment Of The Paper:**

More detailed information regarding each of these aspects is given below:

**Q2(4) Quality Of Experiments (Optional):**

2: Fair: The experimental evaluation is weak: important baselines are missing, or the results do not adequately support the main claims.

**Q2(5) Reproducibility:**

2: Fair: Key resources (e.g., proofs, code, data) are unavailable but key details (e.g., proof sketches, experimental setup) are sufficiently well-described for an expert to confidently reproduce the main results.

**Q3 Main Strengths:**

1. The paper studies an interesting and meaningful problem, DP about multi-party independent data release.
2. The theoretical development for the proposed method.
3. Experiments have shown the promise of the proposed RMGM-OLS algorithm.

**Q4 Main Weakness:**

1. Constraints/restrictions of the proposed method/algorithm.
2. Experiments should be strengthened.


**Q5 Detailed Comments To The Authors:**

*The proposed method/algorithm
1. It's not clear how the privacy constraint regarding B (stated in the last paragraph in column 1, page 3) is assured.
2. The proposed method has made a great start on the problem, but the proposed algorithm is only restricted to the Gaussian mechanism given in Lemma 1 and regression problem, and it is not clear this would impact the applications of the proposed algorithm.

*Experiments
Although the paper addresses that the goal here is to release multi-party datasets (not the final model), it seems to me that  the work done in this paper is to achieve DP and high utility w.r.t. regression, i.e. the focus is still on the final (prediction) model. In this sense, the experiment evaluation should include the work focused on the final model (described in the Related Work section). Moreover, it would be useful to include the straightforward approach (i.e. combining data from all parties in a centralized place) as a baseline, although the paper is restricted to the setting that such combination is not possible.

Additionally, in terms of presentation, it may be better to focus on one algorithm, RMGM-OLS, instead of claiming that two algorithms are proposed and then turning down one algorithm. Later on, perhaps still can use DGM-OLS as a baseline.

**Q7 Justification For Your Score:**

The paper studies a practical problem and the proposed algorithm RMGM-OLS has shown its promise. However, it's not clear whether the constraints and setting would restrict the application of the algorithm, and the evaluation needs to be strengthened too.

**Q9 Complying With Reviewing Instructions:**

1: Yes.

---

### Decision · Program_Chairs · 2022-05-15

**Decision:**

Accept (Poster)

**Comment:**

Meta Review: The paper considers the problem of training linear regression models using disjoint datasets owned by different stakeholders, such that no stakeholder can infer individual level information in datasets they do not own, while the model is asymptotically optimal (in the sense of making use of all the data across all stakeholders).

The authors propose two methods to accomplish this, that they term the de-biased Gaussian mechanism for ordinary least squares (DGM-OLS) and random mixing prior to Gaussian mechanism for ordinary least squares (RMGM-OLS).  The former applies a Gaussian noise corruption when data is released, and subsequently applies a de-biasing operator to the Hessian matrix used in model training.  This approach can be unstable as it involves computing a matrix inverse, which leads to instability in practice.  The latter method aims to avoid the matrix inversion step by introducing noise that is smaller in magnitude, while still maintaining the desired differential privacy properties.

The authors evaluated their method on synthetic and benchmark datasets, and concluded that the RMGM-OLS method generally performed better.

While there were initially some points of confusion, the overall reviewer consensus on this paper was positive after the feedback phase.